# NormCo: Deep Disease Normalization for Biomedical Knowledge Base Construction

**Dustin Wright**                                                        DBW003@ENG.UCSD.EDU

**Yannis Katsis**                                                        YANNIS.KATSIS@IBM.COM

**Raghav Mehta**                                                        R3MEHTA@ENG.UCSD.EDU

**Chun-Nan Hsu**                                                        CHUNNAN@UCSD.EDU

## Abstract

Biomedical knowledge bases are crucial in modern data-driven biomedical sciences, but automated biomedical knowledge base construction remains challenging. In this paper, we consider the problem of disease entity normalization, an essential task in constructing a biomedical knowledge base. We present NormCo, a deep coherence model which considers the semantics of an entity mention, as well as the topical coherence of the mentions within a single document. NormCo models entity mentions using a simple semantic model which composes phrase representations from word embeddings, and treats coherence as a disease concept co-mention sequence using an RNN rather than modeling the joint probability of all concepts in a document, which requires NP-hard inference. To overcome the issue of data sparsity, we used distantly supervised data and synthetic data generated from priors derived from the BioASQ dataset. Our experimental results show that NormCo outperforms state-of-the-art baseline methods on two disease normalization corpora in terms of (1) prediction quality and (2) efficiency, and is at least as performant in terms of accuracy and F1 score on tagged documents.

## 1. Introduction

Modern biomedical sciences are data-driven and depend on reliable databases of biomedical knowledge. These *knowledge bases* are particularly crucial in domains such as precision medicine [Collins and Varmus, 2015], where the idea is to treat patients with the same condition differently according to their genetic profiles. Knowledge bases for precision medicine contain known associations between genetic variants, disease conditions, treatments, and reported outcomes (see, *e.g.*, [Landrum et al., 2017, Forbes et al., 2016]).

In the past decade, natural language processing has advanced in the biomedical domain and has been utilized to automatically extract knowledge from research publications in order to construct knowledge bases. For certain biomedical entities, algorithms are available to provide reliable extraction, for example genes and their protein products [Settles, 2005, Hsu et al., 2008, Wei et al., 2013], diseases [Leaman and Gonzalez, 2008, Dang et al., 2018], and chemicals [Leaman et al., 2015]. However, current NLP algorithms are far from practically useful to cope with the rapid growth of new publications. Meanwhile, AI, NLP, and machine learning in other domains are advancing quickly. It is therefore important to translate these advancements into the biomedical domain.

This paper focuses on the problem of disease normalization, an essential step in the construction of a biomedical knowledge base as diseases are central to biomedicine.The problem has been studied with promising solutions, such as DNorm [Leaman et al., 2013] and TaggerOne [Leaman and Lu, 2016]; however these approaches are based on surface-form feature engineering and shallow

learning methods. The closest problems for which deep learning has been successfully used are entity linking and word sense disambiguation, with deep models that consider context and coherence. However, to the best of our knowledge the problem of how to apply deep learning to solve the problem of disease normalization has not been successfully addressed. Given this, we are concerned with the following questions in this work: How can one design a deep learning model for disease normalization that has high accuracy? How can one train this model given the relative lack of training data and the notorious need for large datasets when training deep models? How efficiently can such a model be trained compared to existing shallow learning approaches?

To address these questions we present NormCo, a deep model designed to tackle issues unique to disease normalization. NormCo makes the following contributions:

- A combination of two sub-models which leverage both semantic features and topical coherence to perform disease normalization.

- Addressing the data sparsity problem by augmenting the relatively small existing disease normalization datasets with two corpora of distantly supervised data, extracted through two different methods from readily available biomedical datasets created for non-normalization-related purposes.

- Outperforming state-of-the-art disease normalization methods in prediction quality when taking into account the severity of errors, while being at least as performant or better in terms of accuracy and F1 score on tagged documents.

- Significantly faster training than existing state-of-the-art approaches (by 2 orders of magnitude faster than the next-best baseline approach, depending on the size of the training dataset).

The paper is structured as follows: We start by describing the problem of disease normalization in Section 2 and surveying related work in disease normalization and entity linking, explaining why disease normalization is unique compared to generic entity linking in Section 3. Sections 4 and 5 present the architecture of our model and its implementation details, respectively. Finally, Section 6 presents the experimental evaluation of NormCo and Section 7 concludes the paper with a discussion of the results and future work toward a fully automated approach to biomedical knowledge base construction.

## 2. Disease Normalization

The primary problem in entity normalization is to map ambiguous spans of text to a unique concept defined in an ontology. The problem can be decomposed into: 1) disambiguating pieces of text which are similar morphologically but should be mapped to different concepts and 2) correctly mapping pieces of text which are different morphologically but refer to the same concept. For example, in Figure 1, a disease mention "Regional enteritis" in an article (left pane) is mapped to a disease concept defined in a given ontology, in this case, the CTD Medic disease dictionary [Davis et al., 2012] (right pane) as "Crohn Disease," with a standard MeSH ID "D003424". The ontology defines a standard vocabulary of a domain. In addition to concept name and ID, an ontology also provides for each concept a definition and a list of synonyms. Disease normalization can be solved by matching a mention to these synonyms, in this case, "Enteritis, Regional" (enlarged pane). Disease normalization can also be solved by looking at the context where a mention appears, and other diseases co-mentioned in the same article ("Crohn's disease" in the abstract, left pane). Co-mentioned

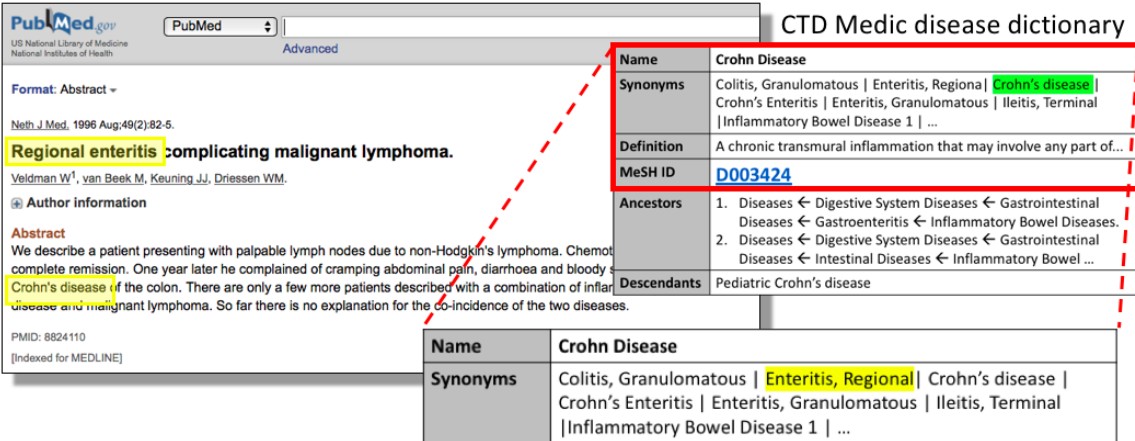

Figure 1: Disease normalization.

diseases are usually related to form a *coherence* topic that disease normalization may potentially leverage. In summary, disease normalization may consider: 1) Similarity of the *mention* text with a known synonym of the target concept, 2) *Context* where the mention appears, and 3) *Coherence* with other co-mentioned diseases.

## 3. Related Work

### 3.1 Disease Normalization

Disease normalization is uniquely challenging due to the evolving conceptualization and nomenclature of diseases. What constitutes a disease may depend on medical, legal, and cultural criteria. A disease concept is intrinsically composite, in the sense that it can be divided into components, *e.g.*, disease site in the body, differentiating signs and symptoms, pathogens that cause the disease, *etc*. Authors may name a disease by combinations of these components. Previous work in disease normalization, therefore, focused mostly on how to search from an incomplete list of synonyms the best match of an input disease mention.

DNorm [Leaman et al., 2013] is a disease normalization method which uses pairwise learning to rank (pLTR) to solve the normalization problem. They first model entity mentions and concept names as TF-IDF vectors. They then learn a similarity matrix which determines a similarity score for a pair of TF-IDF vectors. The model is trained using a margin ranking loss, where mentions from a corpus are matched with associated concept names in the ontology.

TaggerOne [Leaman and Lu, 2016] jointly learns named-entity recognition (NER) and entity normalization using semi-Markov models. TaggerOne considers context during NER and yields better performance than DNorm. The model takes a supervised semantic indexing approach which relies on rich features of the surface forms of input text for NER and TF-IDF weighting for normalization. Similar to DNorm, a similarity matrix is trained to measure the similarity between mention features and concept name features.

More recently, attempts were made to develop deep models to solve the problem of disease normalization. [Li et al., 2017] describes a convolutional neural network model to rank candidate concepts based on word embeddings providing semantic information. The model, however, relies on

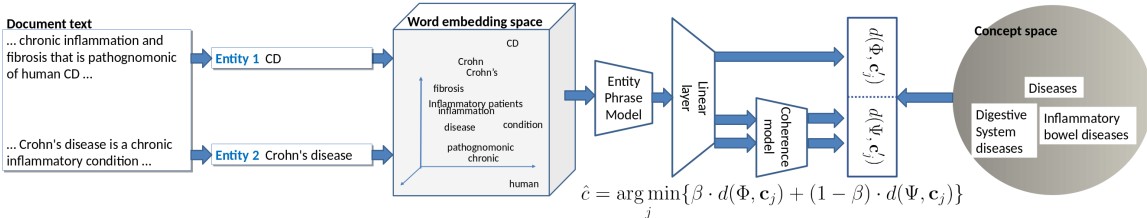

Figure 2: NormCo architecture which utilizes coherence and semantic features for disease normalization.

handcrafted rules to generate candidate concepts. [Cho et al., 2017] describes another method that uses word embeddings as the input to the model. Neither of them considers context or coherence to improve model performance.

## 3.2 Entity Linking

Proposed deep models for disease normalization do not consider mention, context and coherence simultaneously as in recent work on entity linking [Shen et al., 2015]. The entity linking problem is as follows: given a text mention and an ontology of unique entities, determine the entity in the knowledge base to which the text mention refers. Recent methods in entity linking include the use of expectation maximization to train a selective context model [Lazic et al., 2015], knowledge graph densification [Radhakrishnan et al., 2018], and deep neural networks [Ganea and Hofmann, 2017, Sorokin and Gurevych, 2018, Sun et al., 2015, He et al., 2013, Le and Titov, 2018]. Context information has been well-studied as a means to improve normalization performance, where the context can come from the surrounding text in a document or from the knowledge base itself. In addition, the concept of "global" context has been studied in entity linking as a way to induce a coherent set of concepts [Ganea and Hofmann, 2017, Le and Titov, 2018, Ratinov et al., 2011, Hoffart et al., 2011, Ferragina and Scaiella, 2010, Cucerzan, 2007, Milne and Witten, 2008]. In [Lazic et al., 2015], the authors use surrounding referring phrases as context features and selectively pick a single feature for a given mention. They accomplish this by modeling the selected feature as a latent variable conditioned on the mention being linked, drawing all other context features from a background probability distribution. [Ganea and Hofmann, 2017] propose a model which learns to embed both words and entities in a shared space. They exploit local context by scoring words in a context window around the mention text with a set of candidate entities, using the top scoring words and prior probability of an entity given a mention as features in a scoring neural network. Additionally, they use document level context by ensuring coherence among entities using a conditional random field and loopy belief propagation. The model in [Le and Titov, 2018] seeks to exploit latent relationships between $n$ mentions that form a $O(n^2)$-sized clique in order to induce coherence among entities within a document.

## 3.3 Dense Representations of Text

Current methods for representing textual components tend to rely on dense vector representations of the text for performing downstream tasks. These representations can be used to model various aspects of language, such as semantics [Mikolov et al., 2013, Pennington et al., 2014, Peters et al., 2018] or morphology [Ling et al., 2015]. Semantic modeling follows the distributional hypothesis

[Harris, 1954], which proposes that the meaning of a word can be derived from the context in which it occurs. The models of [Mikolov et al., 2013] and [Peters et al., 2018] achieve this by training a neural network to predict context words, while [Pennington et al., 2014] use word co-occurrence statistics as a means to train their word representations. These approaches have been generalized to larger textual components i.e. sentences. [Kiros et al., 2015] use a Gated Recurrent Unit (GRU) network to encode a sentence and decode the surrounding sentences. [Hill et al., 2016] implement a simple bag of words model summing up the word embeddings in a sentence to predict the words in the surrounding sentences. Our entity phrase model is inspired by the simple and effective model of [Hill et al., 2016].

## 4. Methods

### 4.1 Model Overview

Figure 2 shows an overview of the architecture of NormCo, our deep model for disease normalization. Given a document containing a sequence of mentions of disease entities, the model aims at mapping the mention entities to a disease concept space, such that the vector representations of the mentions are close to the vector representations of the disease concepts to which they refer.

In creating this mapping, NormCo utilizes two sub-models; an entity phrase model and a coherence model, each exploiting different aspects of the mentions. The *entity phrase model* leverages the morphological and semantic pattern of a mention, while the *coherence model* exploits other diseases mentioned in the same document. The final model combines both sub-models, which are trained jointly. We will describe each of the two sub-models and explain how they are trained.

To this end, let us first formally define the normalization problem and introduce the notation we will be using in the following sections: Let $D$ be a set of documents, each consisting of a set of entity mentions $M = \{m_0, \ldots, m_K\}$, as well as an ontology $C = \{c_0, \ldots, c_T\}$ of $T$ concepts, where each concept $c_j$ is associated with one or more known names $S_j$. Entity normalization is the problem of determining how to map each mention $m_i \in M$ within a document to a single concept $c_j \in C$ in the ontology (i.e., how to determine the mapping $M \to C$, for each document $d_i \in D$).

### 4.2 Entity Phrase Model

Figure 3a shows the entity phrase model of NormCo. The purpose of this model is to leverage the semantic features of an individual mention in order to map it to a concept. This is accomplished by creating a composite embedding of a mention through summation of the word embeddings of the tokens comprising the mention. Consider a mention $m_i \to c_j$ consisting of tokens $\{w_0, \ldots, w_l\}$. The entity phrase model first embeds the tokens appearing in $m$ into dense vector representations $\{\mathbf{e}_0, \ldots, \mathbf{e}_l\}, \mathbf{e}_i \in \mathbb{R}^d$. For example, the mention "Crohn disease" is initially tokenized into "Crohn" and "disease", and then the word embedding of both tokens is obtained. The phrase representation is the summation of the word embeddings of the individual tokens.

$$\mathbf{e}_i^{(m)} = \sum_{k=0}^{l} \mathbf{e}_k \tag{1}$$

This simple but powerful model has been shown to be useful for learning sentence representations [Hill et al., 2016]. Finally, the composite embedding is passed through a linear projection to obtain

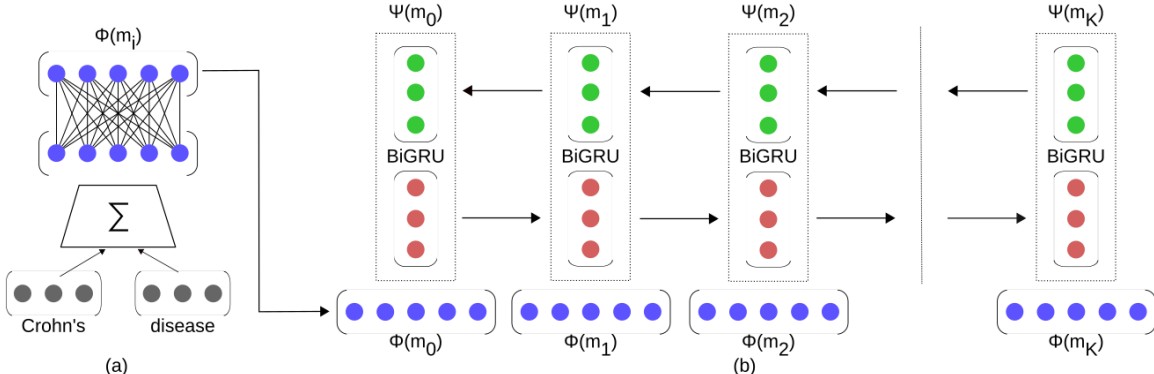

Figure 3: Joint entity-coherence model. (a) The entity phrase model is a simple summation of word embeddings followed by a linear projection. (b) The coherence model takes the entity phrase representation as input and passes a single document of mentions through a bidirectional GRU network.

the entity phrase representation given in Equation 2.

$$\Phi(m_i) = \mathbf{A}\mathbf{e}_i^{(m)} + \mathbf{b} \tag{2}$$

where $\mathbf{A} \in \mathbb{R}^{d \times d}$ and $\mathbf{b} \in \mathbb{R}^d$.

### 4.3 Coherence Model

To model coherence (Figure 3b), we use a bidirectional gated recurrent unit (BiGRU) [Cho et al., 2014] network acting on the conceptual representations of the mentions. The goal of the coherence model is to predict a set of diseases within one document which are likely to appear together. Coherence has been a topic of recent interest in the entity linking community, for example in [Ganea and Hofmann, 2017, Le and Titov, 2018], and has been shown to improve performance on entity linking tasks. Consider a document containing the mentions "irritable bowel syndrome", "ulcerative colitis", and "Crohn disease." The entities surrounding "ulcerative colitis" should influence its output concept mapping such that the set of selected concepts are likely to appear together, thus deterring the model from mapping the entity to nonsensical concepts, such as "Neoplasms".

The purpose of modeling coherence as a sequence is two-fold: 1) the concept representation of a particular phrase becomes a function of the current phrase representation and the surrounding phrase representations and 2) by using a sequence model we can ensure the computational complexity is not NP-hard, which would be the case when modeling the joint probability of the entire set of concepts. Moreover, we used a GRU instead of the commonly used LSTM for sequence modeling based on our initial testing, showing that a GRU yields better performance than an LSTM in our setting. As an additional benefit, a GRU has only a hidden state, unlike an LSTM which has both a hidden state and memory, and thus requires less computation.

In this, we obtain a forward representation $\Psi_f(m_i)$, and a backward representation $\Psi_b(m_i)$, where

$$\Psi_f(m_i) = \text{GRU}(\Phi(m_i)|\Phi(m_{i-1}), ..., \Phi(m_0)) \tag{3}$$

and

$$\Psi_b(m_i) = \text{GRU}(\Phi(m_i)|\Phi(m_{i+1}), ..., \Phi(m_K)) \tag{4}$$

After obtaining this representation for each concept, we concatenate the forward and backward activations of the GRU, yielding the following:

$$\Psi(m_i) = \Psi_f(m_i)||\Psi_b(m_i) \tag{5}$$

where $||$ is the concatenation operation and $\Psi(m_i) \in \mathbb{R}^d$. We chose to concatenate the forward and backward activations of the BiGRU so that the output representation includes features learned from both past and future concepts in the sequence; concatenation is common practice for entity recognition and linking, for example see [Lample et al., 2016].

### 4.4 Training

For each concept $c_j \in C$, we seek to jointly train concept embeddings $\mathbf{c}_j \in \mathbb{R}^d, j \in 0, \ldots, T$ and models $\Phi(m_i)$ and $\Psi(m_i)$ such that the points induced by the phrase and coherence model are closest to their ground truth concept embedding. The reason for jointly training the two models, as opposed to training each model separately, is to ensure that the entity phrase model can learn a good mapping from tokens to concepts, and the coherence model can alter any of the phrase representations as necessary based on the surrounding concepts. Our experimental results, which we discuss in Section 6.4.3, verify that joint training leads to superior performance than separate training of each model.

To train the model, we first define the distance metric used to measure the similarity between $\mathbf{c}_j$ and the vectors $\Phi(m_i)$ and $\Psi(m_i)$ as the Euclidean distance: $d(x, y) = \|x - y\|_2$. We then combine the distances $d(\Phi(m_i), \mathbf{c}_j)$ and $d(\Psi(m_i), \mathbf{c}_j)$ weighted by a learned parameter $\beta$.

$$\hat{d}(m_i, c_j) = \beta \cdot d(\Phi(m_i), \mathbf{c}_j) + (1 - \beta) \cdot d(\Psi(m_i), \mathbf{c}_j) \tag{6}$$

The purpose of $\beta$ is to automatically learn how to combine the score for each model. When a particular mention $m_i$ maps to a concept $c_j$, we wish to train the model to minimize $\hat{d}(m_i, c_j)$. We accomplish this by using a max-margin ranking loss with negative sampling:

$$\mathcal{L} = \frac{1}{|N|} \sum_{k \in N} \max\{0, P + \hat{d}(m_i, c_j) - \hat{d}(m_i, c_k)\} \tag{7}$$

where $c_k$ is a negative example of a concept, $P$ is the margin, and $N$ is the set of negative examples. In this, the model is encouraged to minimize the distance between the ground truth concept and the combined phrase distances, while maximizing its distance to negative examples. In practice, we found it useful to first pass $\Phi(m_i)$ and $\Psi(m_i)$ through the logistic sigmoid function before calculating the distance and setting the margin $P$ to $\sqrt{d}$, yielding the following[1].

$$\mathcal{L}_s = \frac{1}{|N|} \sum_{k \in N} \max\{0, \sqrt{d} + \hat{d}_s(m_i, c_j) - \hat{d}_s(m_i, c_k)\}, \tag{8}$$

where $\hat{d}_s$ is defined as

$$\hat{d}_s(m_i, c_j) = \beta \cdot d(\sigma(\Phi(m_i)), \sigma(\mathbf{c}_j)) + (1 - \beta) \cdot d(\sigma(\Psi(m_i)), \sigma(\mathbf{c}_j)). \tag{9}$$

---

1. https://towardsdatascience.com/lossless-triplet-loss-7e932f990b24

We utilize both the entity phrase model and coherence model by jointly training both networks. The entity phrase model is trained using both the mentions $M$ from the training corpus as well as all of the known names $S_j \in C$ from the given ontology. In this, we train the model to map each $m_i \in M$ and $s_k \in S_j$ to its corresponding concept $c_j$. The coherence model makes use of the supervised training data as well as distantly supervised data.

### 4.5 Inference

We process annotations one document at a time, passing each annotation $m_i$ through the entity phrase and coherence models. We then match each vector $\Phi(m_i)$ and $\Psi(m_i)$ to one of the concept vectors in the ontology using the distance metric defined in Equation 9. The selected concept $\hat{c}_i$ for mention $m_i$ is then determined using Equation 10.

$$\hat{c}_i = \underset{c_j \in C}{\arg\min}\{\hat{d}_s(m_i, c_j)\}. \tag{10}$$

### 4.6 Data Augmentation

Datasets for biomedical entity normalization [Doğan et al., 2014, Wei et al., 2015] tend to be relatively small in terms of the number of training examples (i.e. 500 abstracts in [Doğan et al., 2014]), while training deep neural networks usually requires substantial amounts of training examples in order to be effective. In order to compensate for a lack of examples of co-occurring diseases, which is necessary to train our models, we generated two corpora of additional training data by leveraging existing labeled datasets created for other non-normalization-related tasks.

The first corpus is a corpus of *distantly supervised data*, obtained by leveraging the BioASQ dataset [Tsatsaronis et al., 2015], a large scale biomedical semantic indexing corpus. BioASQ contains over 23GB of PubMed abstracts labeled by humans with MeSH terms, corresponding to the main biomedical entities (including diseases) mentioned in the paper. While readily available, MeSH term annotations are provided at the paper level and not associated with any disease mention in the abstract. As a result, they cannot be directly used as ground truth labels. To generate high quality labels for disease mentions out of the MeSH terms we use the following criteria 1) A disease mention exists in the text which matches one of the associated names of the MeSH terms listed with the paper; 2)Less than 20 and more than 4 disease mentions are found in the given abstract; 3) At least 80% of the MeSH terms listed with the paper have a corresponding disease mention which can be found using the ontology.

The second corpus is a corpus of *synthetic data* generated by using data driven co-occurrence probabilities derived from the BioASQ dataset. We estimate these probabilities by counting the co-occurrences of concepts based on the list of MeSH terms given with each abstract. Since the dataset covers approximately 40% of the concepts in the ontology, we estimate the unseen concepts by replacing their count with that of the nearest parent in the concept poly-hierarchy with the lowest unary occurrence. The reason for choosing concepts with the lowest occurrence is that such concepts tend to be more specifically related to the target concept. For example, consider "Teeth injury" as our target with zero observations. "Teeth injury" has two parents, "Teeth disorders" and "Wounds and injuries." They are equally close to "Teeth injury" but we chose the less-observed "Teeth disorders" rather than "Wounds and injuries." Conceptually, "Teeth disorders" is more specifically related to "Teeth injury" than "Wounds and injuries". Using the co-occurence probabilites calculated from the BioASQ dataset, we randomly sample sets of concepts. For each set of concepts, we then randomly

sample an associated synonym for each concept during each batch of training. As we will discuss in our experimental evaluation in Section 6.4.4, the data augmentation described above is crucial for the high predictive accuracy of our model.

## 5. Implementation[2]

For our experiments, we used the Comparative Toxicogenomics Database (CTD) MEDIC dictionary [Davis et al., 2012] as the concept ontology. This ontology maps disease concepts to a particular MeSH (Medical Subject Headings) [Lipscomb, 2000] or OMIM (Online Mendelian Inheritance in Man) [Hamosh et al., 2005] identifier. Each concept is equipped with a preferred name, identifier, and zero or more synonyms. In order to utilize the concept names listed in the ontology, we start each epoch by passing batches of mention only examples through the entity phrase model and directly optimizing $\mathcal{L}$ with $\beta$ set to 1.0. This yields an additional 79,248 training examples for the entity phrase model, and allows the model to learn about concepts which are rarely or never seen in the training corpus.

All of the models were implemented using PyTorch. We used $d = 200$ dimensional word2vec embeddings pre-trained on PubMed abstracts and PubMed Central articles to initialize our word embeddings [Mikolov et al., 2013, Moen and Ananiadou, 2013]. Additionally, we initialized the concept embeddings by taking a summation of word embeddings of the tokens in the preferred name for each concept. For text preprocessing, we remove all punctuation from the mention, lowercase, and tokenize using the NLTK [Bird and Loper, 2004] Treebank word tokenizer. Following previous work [Leaman et al., 2013, Leaman and Lu, 2016] we resolve abbreviations found in the text, mapping them to their long form, as a preprocessing step [Sohn et al., 2008]. All token sequences are truncated to a length of at most 20. We train in batches of 10 when training only mentions, and single batches containing the mentions of one abstract when training the full model. Negative samples are randomly selected among all of the concepts in the ontology, with the number of samples set to $N = 30$. To optimize the network we use the PyTorch implementation of the Adam optimizer [Kingma and Ba, 2014] with the initial learning rate set to 0.0005. We use a validation set in order to benchmark the generalization performance of the model during training, as well as to perform early stopping. Our models tend to converge within 10-30 epochs.

## 6. Experimental Evaluation

### 6.1 Datasets

We evaluate the system using two entity normalization corpora from the biomedical domain: the NCBI disease corpus and the BioCreative V Chemical/Disease Relation corpus. The NCBI disease corpus [Doğan et al., 2014] consists of scientific abstracts available on PubMed labeled with disease mentions. It is split into a training set of 500 abstracts and a development and test set of 100 abstracts each. Each disease mention is labeled with either its MeSH or OMIM identifier, defined in the MEDIC Disease Dictionary [Davis et al., 2012]. Additionally, some mentions (*e.g.*, "breast and ovary cancers") are labeled as composite mentions and can refer to two or more different concepts in the ontology. We treat these examples as multiple training examples, training the same mention to map to each of the possible concepts. We only map to a single concept during inference. The

---

2. https://github.com/IBM/aihn-ucsd/tree/master/NormCo-deep-disease-normalization

BioCreative V Chemical/Disease Relation (BC5CDR) corpus [Wei et al., 2015] consists of 1500 abstracts (1000 train, 500 test) gathered from PubMed, and similarly contains disease mentions labeled with their MeSH/OMIM identifiers with some composite mentions. More details on the datasets are given in Table 1.

| Corpus | Annotations | Mentions | Concepts |
|---|---|---|---|
| NCBI Disease | 6892 | 2135 | 753 |
| BC5CDR (Disease) | 12864 | 3271 | 1082 |

Table 1: Annotations, unique mentions, and unique concepts present in each dataset.

We split the BioCreative V training set into a train and validation set of 950 and 50 abstracts respectively. Using our training data collection method, we obtain an additional 13,665 labeled abstracts. For the NCBI disease corpus, we use the original train/test/dev split and obtain an additional 13,193 abstracts for distant supervision. Additionally, for the BioCreative V task we add a unique concept to the ontology labeled "⟨unk⟩" for mentions that should return a NIL result.

## 6.2 Baseline Methods and Models

As baseline methods, we used DNorm [Leaman et al., 2013] and TaggerOne [Leaman and Lu, 2016] which correspond to the current state-of-the art for disease normalization on both the NCBI disease corpus and BioCreative V datasets. To study the effects of various aspects of NormCo, we created and compared the following versions of our model.

- **Mention only (MO)**: Includes only the entity phrase model ($\Phi(m_i)$) trained on the mentions in the corpus and the synonyms in the ontology.

- **Mention + Coherence (MC)**: Includes both the phrase model and coherence model trained on the mentions in the corpus and the synonyms in the ontology.

- **Distantly supervised data (\*-distant)**: The given model trained using the dataset, dictionary data, and distantly supervised data.

- **Synthetic data (\*-synthetic)**: The given model trained on synthetically generated data using the prior probability described in the previous section, as well as distantly supervised data, dictionary data, and the dataset.

We also note that both of the state of the art baseline methods require training a large similarity matrix, the size of which is dependent on the size of the vocabulary due to the use of TF-IDF vectors. In contrast, our model leverages dense word embeddings, which allow it to capture more compact representations of the text and require 2 orders of magnitude less parameters (Table 2).

| Model | DNorm | TaggerOne | **NormCo** |
|---|---|---|---|
| # Parameters | 397,444,096 | 500,392,260 | 9,151,801 |

Table 2: Number of trainable parameters in each model.

### 6.3 Experimental Metrics

#### 6.3.1 MIRCO-F1 AND ACCURACY

We first evaluated the performance of our approach using micro-F1 and perfect tagger accuracy. For micro-F1 experiments we used the NER tags output by TaggerOne. Following the approach in [Leaman et al., 2013] and [Leaman and Lu, 2016], our micro-F1 score is calculated at the instance level. That is, an annotation is considered a true positive if a correct concept is found within a particular abstract, ignoring span boundaries. For example, if the concept for "Crohn disease" appears in abstract $A$ and the model determines that a mention in abstract $A$ maps to Crohn disease, regardless of where the mention appears, it is considered a true positive. In the micro-F1 setting, our results are averaged across the entire set of documents considered. The reason for using micro-F1 (in contrast to macro-F1) is to allow comparison to evaluation metrics reported by recent literature in the area.

For perfect tagger accuracy, we tested the ability of each system to normalize the exact mention text from the gold standard test set. For the results labeled "AwA," we disabled abbreviation resolution for each method during testing in order to understand which normalization method performs better for a predetermined set of mentions, as well as the contribution of the abbreviation resolution module. For Acc@1, we tested the full pipeline for each method using a perfect tagger and obtained top 1 normalization accuracy.

#### 6.3.2 NORMALIZED LOWEST COMMON ANCESTOR DISTANCE

While accuracy gives a strong indicator of basic performance, it does not take into account the overall quality of the predictions made by a normalization system. In particular, accuracy treats every error equally regardless of the severity of the error (i.e., how far it is from the ground truth). To understand prediction quality, we evaluated each model using normalized lowest common ancestor (LCA) distance, a metric which utilizes the concept ontology to determine how close each prediction is to the ground truth. We first define an operator $\text{LCA}(c_1, c_2) = d(c_1, c) + d(c_2, c)$ which quantifies the distance between a predicted concept $c_1$ and a ground truth concept $c_2$ as the sum of the path distances of $c_1$ and $c_2$, respectively from their lowest common ancestor $c$. Then we leverage this operator to compute the average LCA distance between each pair of predicted and ground truth concepts as follows:

$$\bar{d}_{\text{LCA}} = \frac{\sum_{i \in I} \text{LCA}(\hat{c}_i, c_{gt})}{|I|}$$

where $I$ is the set of all mentions across all documents. This provides a measure which takes into account both accuracy (the predictions which have a distance of 0) and severity of error (the distance of errors to the ground truth), normalizing over both.

#### 6.3.3 TRAINING TIME

Finally, we use training time as a way to compare the performance of each model. For these experiments, we observe the total time per epoch required to train the model.

| (a) NCBI Disease Corpus | | | | | | |
|---|---|---|---|---|---|---|
| Model | P | R | F1 | AwA | Acc@1 | $\bar{d}_{\mathrm{LCA}}$ |
| DNorm | 0.803 | 0.763 | 0.782 | 0.778 | 0.872 | 0.408 |
| TaggerOne | 0.822 | 0.792 | 0.807 | 0.811 | 0.877 | 0.389 |
| MO | 0.846 | 0.801 | 0.823 | 0.817 | 0.860 | 0.389 |
| MC | 0.843 | 0.799 | 0.820 | 0.818 | 0.865 | **0.381** |
| MO-distant | 0.855 | 0.811 | 0.833 | 0.827 | 0.870 | *0.388* |
| MO-synthetic | 0.843 | 0.800 | 0.821 | 0.807 | 0.855 | 0.443 |
| MC-distant | 0.839 | 0.796 | 0.817 | 0.819 | 0.856 | 0.465 |
| MC-synthetic | **0.863** | **0.818** | **0.840** | **0.828** | **0.878** | *0.387* |
| (b) BC5CDR Disease Dataset | | | | | | |
| Model | P | R | F1 | AwA | Acc@1 | $\bar{d}_{\mathrm{LCA}}$ |
| DNorm | 0.812 | 0.801 | 0.806 | 0.838 | 0.879 | 0.444 |
| TaggerOne | 0.846 | **0.827** | **0.837** | 0.852 | **0.889** | 0.450 |
| MO | 0.854 | 0.794 | 0.823 | 0.840 | 0.860 | 0.451 |
| MC | 0.861 | 0.800 | 0.830 | 0.850 | 0.868 | *0.431* |
| MO-distant | 0.861 | 0.800 | 0.830 | 0.851 | 0.875 | 0.449 |
| MO-synthetic | 0.862 | 0.802 | 0.831 | 0.854 | 0.876 | **0.412** |
| MC-distant | **0.866** | 0.805 | 0.834 | **0.857** | 0.880 | *0.434* |
| MC-synthetic | 0.862 | 0.801 | 0.830 | 0.853 | 0.874 | *0.420* |

Table 3: Comparison of results for (a) NCBI and (b) BC5CDR Disease datasets. AwA is accuracy with abbreviation resolution disabled for all models. Acc@1 is perfect tagger accuracy. $\bar{d}_{\mathrm{LCA}}$ is the normalized LCA distance (lower is better). Bold numbers indicate best performance.

## 6.4 Results and Discussion

### 6.4.1 MICRO-F1, ACCURACY, AND NORMALIZED LCA DISTANCE

Our results on each dataset are given in Table 3. For micro-F1, our models were able to consistently improve over the precision of the baseline methods while maintaining good recall. On the NCBI disease corpus, our coherence model with distantly supervised and synthetic data outperformed [Leaman and Lu, 2016] in terms of F1 by 0.033. The largest gain was in terms of precision, which improved by 0.041 for NCBI diseases and 0.02 for BioCreative diseases. In terms of accuracy, NormCo maintains good performance when the abbreviation resolution module is removed, suggesting that it is better able to understand abbreviations than the baseline methods. Our overall accuracy with the entire pipeline outperforms the baselines on the NCBI disease corpus and is highly competitive at only 0.9% under the strongest baseline for BioCreative diseases. The slightly lower F1 of the model on the BC5CDR dataset is most likely due to the model prioritizing precision over recall, thus performing better on the NCBI dataset that has a smaller set of diseases compared to the more diverse set of concepts appearing in BC5DR (753 vs 1082).

Additionally, our models consistently produce higher quality predictions than the baseline methods in terms of the average distance to common ancestors. This indicates that when a mention is mislabelled according to the ground truth, our predictions tend to be more related to the original concept than the baseline methods. For example, on the NCBI disease corpus 85.5% of the errors

made by our best model (MC-synthetic) are either a parent or child concept of the ground truth concept (compared to 72.9% for DNorm and 81% for TaggerOne). This is evident in behavior such as the mention "congenital eye malformations," which has a ground truth label of "Eye Abnormalities" (MESH:D005124), being mapped to "Congenital Abnormalities" (MESH:D000013), a parent concept of the ground truth. This is a less egregious error than, say, if it were mapped to "Crohn Disease" (MESH:D003424).

### 6.4.2 EFFECT OF COHERENCE

We noticed only a slight improvement of the model when coherence was added. There are a few cases we saw where the model appeared to be utilizing coherence to assist with normalization (Table 4).

| PubMed ID | Mention Text | MO | MC (Correct) | Disease Context |
|-----------|--------------|-----|--------------|-----------------|
| 18189308 | cystitis | Hemorrhage | Cystitis | bladder inflammation; urinary bladder inflammation |
| 23433219 | psychotic symptoms | Psychotic disorders | Psychoses, Substance-induced | depressive disorder, bipolar disorder |
| 10390729 | hypoactivity | Dyskinesia | Movement disorders | hyperactivity |

Table 4: Examples where the coherence model correctly identifies a disease but mention only fails.

In these cases, the model with coherence learns the correct concept mapping using the surrounding diseases, whereas the mention only model, having only the mention text to use for normalization, incorrectly identifies the diseases. For example, the mention only model incorrectly learns to normalize "cystitis" to "Hemorrhage" instead of "Cystitis". This is likely caused by several examples in the training data where the mention "hemorrhagic cystitis" has a composite label containing both the concept for "Hemorrhage" and the concept for "Cystitis." The model with coherence utilizes the surrounding mentions of "bladder inflammation" and "urinary bladder inflammation" during training to help identify the concept as "Cystitis," which is directly related to the urinary tract. However, while the model has improved performance with coherence present, we noticed it tends to learn to use mention only ($\beta$ approaches 1.0) during inference. We conclude that training with coherence influences how the model learns to map mentions to concepts, ultimately relying on the mention model. We plan to further study the utility of coherence for normalization in our future work.

### 6.4.3 JOINT TRAINING VS. SEPARATE TRAINING

We next observed the effects of joint vs separate training of the two models. The results for training with the three different collection types on the two datasets are given in Tables 5. Overall, the model tended to perform better when trained jointly, with the distance combined using the learned parameter $\beta$.

### 6.4.4 DATA AUGMENTATION

We studied the effect of augmenting the dataset with terms derived from the ontology (Table 6), as well as the impact of distantly supervised data (Figure 4a). We first performed an experiment in which we removed all of the dictionary mentions ($S_i$) and distantly supervised mentions from the training data. The effect of this was a drop in the performance of the model by over 12%. We

| Model | NCBI | | | | BC5CDR | | | |
|---|---|---|---|---|---|---|---|---|
| | Joint | | Separate | | Joint | | Separate | |
| | F1 | Acc@1 | F1 | Acc@1 | F1 | Acc@1 | F1 | Acc@1 |
| MC | 0.820 | 0.865 | 0.825 | 0.857 | 0.830 | 0.868 | 0.826 | 0.865 |
| MC-distant | 0.833 | 0.870 | 0.805 | 0.837 | 0.834 | 0.880 | 0.820 | 0.865 |
| MC-synthetic | 0.840 | 0.878 | 0.781 | 0.808 | 0.830 | 0.874 | 0.826 | 0.867 |

Table 5: Joint vs Separate training of coherence and mentions on NCBI and BC5CDR datasets.

observed that is was critically important to train with both the supervised mentions as well as the synonyms from the ontology. Without the larger training set the model is unable to generalize well to unseen data.

We next investigated how distantly supervised data contributed to the overall performance of the system. Figure 4a shows the accuracy of the MC model versus the amount of supervised training data used. The model was evaluated on the BC5CDR dataset. At 0% we only presented the model with distantly supervised data and the names from the ontology. The model with no supervised training data already achieved over 76% accuracy, which was close to the model trained with only supervised data. Then, as supervised training data were added, the accuracy of the model improved in an approximately linear relationship to the amount of supervised training data provided. This shows that the model was able to learn from both the supervised and distantly supervised data, as it is the combination of these two datasets (as opposed to each dataset individually) that allows the model to achieve high perfomance.

| Dataset | P | R | F1 | Acc@1 |
|---|---|---|---|---|
| NCBI | 0.743 | 0.705 | 0.723 | 0.723 |
| BC5CDR | 0.785 | 0.730 | 0.756 | 0.774 |

Table 6: Results when only coherence data is used for training (no dictionary synonyms or distant supervision).

### 6.4.5 RUNTIME PERFORMANCE

Finally, we looked at the runtime performance of each of the models. We measured the total time per training epoch of each model using only the supervised training data. All experiments were run on the same machine with 3.2GHz 6-core 8th-generation Intel Core i7 and 16GB 2666MHz DDR4 RAM. We scaled the number of training samples i.e. number of articles, from 10% to 100% in intervals of 10% performing 5 runs at each setting and considering the average. All of the models scale linearly with the size of the training set. TaggerOne was the most demanding model of all three, eventually saturating all of the memory. DNorm, while efficient and less memory hungry than TaggerOne, still took about 6 minutes per epoch to train when using half of the training set and about 10 minutes per epoch for the whole set. NormCo outperforms DNorm by taking 40 seconds per epoch when using half of the training set and around 85 seconds per epoch when using the whole set. Overall we observed a 7.88x-15x speedup compared to DNorm and a 2-3 orders of magnitude improvement over TaggerOne, making NormCo significantly faster to train than both baseline approaches.

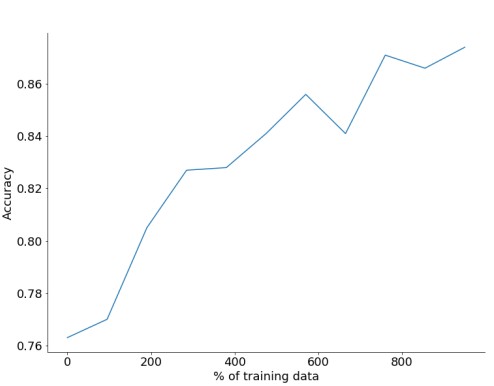

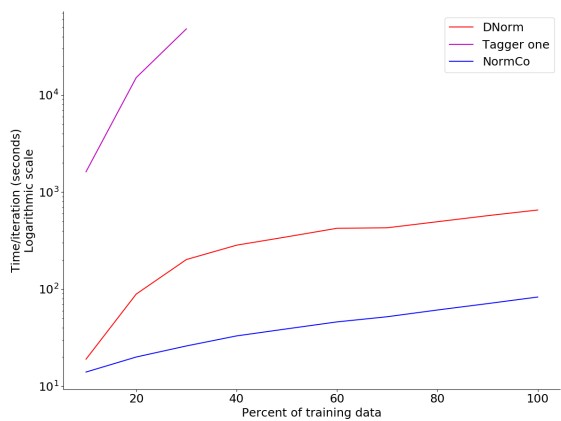

(a) Test set accuracy in relation to the amount of supervised training data used on the BioCreative dataset. At 0%, the model was only presented with distantly supervised data and dictionary synonyms. In this setting the model already achieved 76% accuracy on the test set.

(b) Time (seconds) per epoch of training using subsets of the training samples. Averaged across 5 runs at each setting. Used original source code from authors.

Figure 4: Additional performance comparisons.

## 7. Conclusion

Though deep models perform well in many domains, there is less evidence of them being utilized well in the acceleration and improvement of the construction of biomedical knowledge bases. This is due to the unique challenges in the biomedical domain, including a lack of a large training corpora. In this paper, we propose a deep model which considers semantics and coherence simultaneously to solve the problem of disease normalization, an essential step in the automated construction of biomedical knowledge bases. We demonstrate that the MeSH ontology and BioASQ dataset can be used as a useful source of additional data to support the training of a deep model to achieve good performance. Finally, we show that a model based on semantic features and coherence provides higher quality predictions for the task of disease normalization over state-of-the-art solutions. Still, many uniquely challenging AI problems await to be solved before fully automated construction of biomedical knowledge bases is possible. These problems deserve more attention and investment from the AI research community.

### 7.1 Acknowledgment

This work is supported by IBM Research AI through the AI Horizons Network.

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
