# OpenReview forum: "NormCo: Deep Disease Normalization for Biomedical Knowledge Base Construction"
_AKBC.ws/2019/Conference — AKBC 2019_

### Official Review · AnonReviewer3 · 2019-01-03
**very competent work on an important problem**

**Rating:** 8
**Confidence:** 4

**Review:**

The paper presents a method for named entity disambiguation
tailored to the important case of medical entities,
specifically diseases with MeSH and OMIM
as the canonicalized entity repository.
The method, coined NormCo, is based on a cleverly designed
neural network with distant supervision from MeSH tags of
PubMed abstracts and an additional heuristic for estimating
co-occurrence frequencies for long-tail entities.

This is very competent work on an important and challenging
problem. The method is presented clearly, so it would be easy
to reproduce its findings and adopt the method for further
research in this area.
Overall a very good paper.

Some minor comments:

1) The paper's statement that coherence models have
been introduced only recently is exaggerated.
For general-purpose named entity disambiguation, coherence
has been prominently used already by the works of
Ratinov et al. (ACL 2011), Hoffart et al. (EMNLP 2011)
and Ferragina et al. (CIKM 2010); and the classical
works of Cucerzan (EMNLP 2007) and Milne/Witten (CIKM 2008)
implicitly included considerations on coherence as well.
This does not reduce the merits of the current paper,
but should be properly stated when discussing prior works.

2) The experiments report only micro-F1. Why is macro-F1
(averaged over all documents) not considered at all?
Wouldn't this better reflect particularly difficult cases
with texts that mention only long-tail entities,
or with unusual combinations of entities?

---

> ### Author Response · Authors · 2019-02-01
> **Thank you for the feedback**
>
> Thank you for the review and constructive feedback. We address the raised concerns below:
>
> (1) Thank you for the comment. In the updated submission we revised the related work section to better represent existing work on coherence, including the list of works mentioned above.
>
> (2) We agree that macro-F1 could give valuable insights into the performance of the normalization algorithms on rarely seen entities. The reason we reported micro-F1 numbers was mainly to keep in line with the recent publications in the area on the particular datasets that focus on micro-F1 performance. However, we still evaluated the macro-F1 performance of our model on NCBI and found out that it outperformed DNorm with 0.856/0.823/0.833 P/R/F1 compared to 0.828/0.809/0.819 P/R/F1 for DNorm (TaggerOne does not report macro-F1). We added a brief discussion of this to Section 6.3.1.

---

### Official Review · AnonReviewer2 · 2019-01-06
**Simple, fast method with decent results on disease normalization (linking)**

**Rating:** 7
**Confidence:** 5

**Review:**

Summary:
The authors address the problem of disease normalization (i.e. linking). They propose a neural model with submodules for mention similarity and for entity coherence. They also propose methods for generating additional training data. Overall the paper is nicely written with nice results from simple, efficient methods.

Pros:
- Paper is nicely written with good coverage of related work
- LCA analysis is a useful metric for severity of errors
- strong results on the NCBI corpus
- methods are significantly faster and require far fewer parameters than TaggerOne while yielding comparable results

Cons:
- Results on BC5 are mixed. Why?
- Data augmentation not applied to baselines
- Methods are not very novel

Questions:
- Were the AwA results applied only at test time or were the models (including baselines) re-trained using un-resolved abbreviation training data?

---

> ### Author Response · Authors · 2019-02-01
> **Thank you for the feedback**
>
> Thank you for the review! We address some of the issues raised below:
>
> (1) The reason that results on BC5 are mixed is that our model is more conservative, favoring high precision over recall (see Table 3). Since the BC5CDR dataset has a greater diversity of concepts that the NCBI dataset (1082 concepts in BC5CDR compared to 753 in NCBI), the lower recall becomes more important, leading to a slightly lower accuracy than the baseline models. However, note that even in this case the NormCo model still outperforms the baselines on the average LCA distance performance metric, which, as explained in the paper, takes into account not only the overall accuracy but also the severity of the errors. We added an explanation of this to Section 6.4.1.
>
> (2) The AwA models were applied only at test time and the models were not re-trained. We have added language to make this clearer in Section 6.3.1. The purpose of this experiment was to observe how abbreviation resolution affects the performance of the trained models.

---

### Official Review · AnonReviewer1 · 2019-01-08
**Very interesting paper that describes a positive contribution to the state of the art in BioNLP**

**Rating:** 9
**Confidence:** 4

**Review:**

This paper proposes a deep-learning-based method to solve the known BioNLP task of disease normalization on the NCBI disease benchmark (where disease named entities are normalized/disambiguated against the MeSH and OMIM disease controlled vocabularies and taxonomies). The best known methods (DNorm, TaggerOne) are based on a pipeline combination of sequence models (conditional random fields) for disease recognition, and (re)ranking models for linking/normalization.

The current paper proposes instead an end-to-end entity recognition and normalization system relying on word-embeddings, siamese architecture and recursive neural networks to improve significantly (4%, 84 vs 80% F1-score, T. 3). A key feature is the use of a GRU autoencoder to encode or represent the "context" (related entities of a given disease within the span of a sentence), as a way of approximating or simulating collective normalization (in graph-based entity linking methods), which they term "coherence model". This model is combined (weighted linear combination) with a model of the entity itself.
Finally, the complete model is trained to maximize similarity between MeSH/OMIM and this combined representation.
The model is enriched further with additional techniques (e.g., distant supervision).

The paper is well written, generally speaking. The evaluation is exhaustive. In addition to the NCBI corpus, the BioCreative5 CDR (chemical-disease relationship) corpus is used. Ablation tests are carried out to test for the contribution of each module to global performance. Examples are discussed.

There are a few minor issues that it would help to clarify:

(1) Why GRU cells instead of LSTM cells?
(2) Could you please explain/recall why (as implied) traditional models are NP-hard? I didn't get it. Do you refer to the theoretical complexity of Markov random fields/probabilistic graphical models? Maybe you should speak of combinatorial explosion instead and give some combinatorial figure (and link this to traditional methods). My guess is that this is important, as the gain in runtime performance (e.g., training time - F. 4) might be linked to this.
(3) A link should be added to the GitHub repository archiving the model/code, to ensure reproducibility of results.
(4) Could you please check for *statistical significance* for T. 3, 5, 6 and 7? At least for the full system (before ablations). You could use cross-validation.

---

> ### Author Response · Authors · 2019-02-01
> **Thank you for the feedback**
>
> Thank you for the review and insightful questions. We address them below:
>
> (1) In general, based on anecdotal evidence it seems that the relative predictive performance of GRU and LSTM cells depends on the particular task at hand. In our case, we selected GRU cells based on experiments we performed with both LSTM and GRU cells during the model design, which showed that GRU cells led to better results. Another potential benefit of this choice is increased training performance, as GRU cells are less complex and less computation-intensive than LSTM cells. We revised Section 4.3 to explain the reasoning behind our choice of GRU cells.
>
> (2) We are referring to the complexity of modeling and performing inference from the joint probability of the entire set of tags, which is an NP-hard problem. To avoid the exponential blowup, existing techniques employ different types of approximation algorithms (e.g., Ganea and Hoffman (2017) present an N^2 approximation algorithm using a fully-connected pairwise conditional random field, which requires loopy belief propagation to train). Our proposal is to model the problem as a tag sequence using a recurrent net to avoid combinatorial explosion, though other solutions could also be proposed to reduce the complexity (i.e. model it as a tag sequence and use a conditional random field or a hidden Markov model). We cleaned up the language surrounding this point both in the abstract and in Section 4.3.
>
> (3) We intend to make the code of the best models for each dataset available upon acceptance and will be providing a link to it in the paper.
>
> (4) We attempted to obtain significance results during the author feedback period by performing 10-fold cross-validation on the NCBI disease corpus both for the best NormCo model and the best baseline model (which is TaggerOne). While we were able to obtain the evaluation metrics for NormCo, we ran into several issues while retraining TaggerOne on new splits, including (a) TaggerOne’s code breaking (i.e., throwing null pointer exceptions) and (b) TaggerOne’s internal F-score evaluation failing for concepts that have multiple labels (such as “inherited neuromuscular disease”, which is mapped to both MESH:D009468 “Neuromuscular disease” and MESH:D030342 “Genetic diseases, inborn”), which were not present in the original test set. Ultimately these issues, coupled with TaggerOne’s long training times documented in Section 6.4.5, did not allow us to obtain significance results for TaggerOne. However, we were able to perform cross-validation of the best NormCo model (i.e., MC-synthetic), which resulted in an average accuracy of 0.853 with a low standard deviation of 0.013.

---

### Comment · AnonReviewer1 · 2022-04-16
**Very interesting paper that describes a positive contribution to the state of the art in BioNLP**

I hope this paper was published.

---

### Meta-Review · Area_Chair1 · 2019-02-11
**Consensus accept; reviewer concerns addressed in revisions**

**Recommendation:** Accept (Poster)
**Confidence:** 5

**Metareview:**

The reviewers all agree the paper is a clear accept.  The paper presents an end-to-end approach to biomedical concept normalization that supplants previous state of the art pipeline systems based on more conventional bio NLP methods.  Although the individual components of the solution are not novel, e.g.,  siamese networks, GRUs, and distant supervision, etc.,  they are combined together in highly appropriate ways to solve a difficult entity linking problem.  The authors did a commendable job addressing the reviewers comments, questions and concerns by running experiments, providing new results, updating related work to more accurately capture the fact that other entity linking approaches also capture coherence, and addressing a few minor clarity issues.

---

### Decision · Program_Chairs · 2019-02-15
**AKBC 2019 Conference Decision**

Accept